# Friction Stir Processing on the Tribological, Corrosion, and Erosion Properties of Steel: A Review

Alessandro M. Ralls [ID], Ashish K. Kasar [ID] and Pradeep L. Menezes *[ID]

Department of Mechanical Engineering, University of Nevada, Reno, NV 89557, USA;
alessandroralls@nevada.unr.edu (A.M.R.); akasar@nevada.unr.edu (A.K.K.)
* Correspondence: pmenezes@unr.edu

**Abstract:** The eventual material degradation of steel components in bio-implant, marine, and high-temperature applications is a critical issue that can have widespread negative ramifications from a safety and economic point of view. Stemming from their tribological, corrosion, and erosion-based properties, there is an increasing need to address these issues effectively. As one solution, surface processing techniques have been proposed to improve these properties. However, common techniques tend to suffer from issues spanning from their practicality to their high costs and negative environmental impacts. To address these issues, friction-stir-processing (FSP) has been one technique that has been increasingly utilized due to its cost effective, non-polluting nature. By inducing large amounts of strain and plastic deformation, dynamic recrystallization occurs which can largely influence the tribological, corrosion, and erosion properties via surface hardening, grain refinement, and improvement to passive layer formation. This review aims to accumulate the current knowledge of steel FSP and to breakdown the key factors which enable its metallurgical improvement. Having this understanding, a thorough analysis of these processing variables in relation to their tribological, corrosion, and erosion properties is presented. We finally then prospect future directions for this research with suggestions on how this research can continue to expand.

**Keywords:** friction stir processing; steel; dynamic recrystallization; wear; corrosion; erosion

## 1. Introduction

Friction stir processing (FSP) is a material processing technique developed in 1999 derived from friction stir welding (FSW). Developed by Mishra et al. [1,2], this process utilizes localized plastic deformation by rotating a specialized pin through the working piece [3], as shown in Figure 1. Typically, these specialized pins tend to vary in geometry depending on the experiment, whether optimizing material flow or increasing the frictional heat and plastic deformation along the processed region [4]. These pins are selected based on various characteristics, which can broadly impact the final surface finish [5,6]. When considering a pin, the selected material should have a suitable wear resistance against the working piece. For example, the suitable pin materials for FSP on steel are Tungsten and Molybdenum. This is crucial in order to sustain the long-term usage of the tooling piece. Of course, this implies that the tooling piece will have a combination of high toughness and strength for a variety of temperature ranges with the working temperature of the pin being within the range of 60–80% of the absolute melting temperature [7]. Considering these points, the thermal fatigue strength should be optimized while maintaining a low thermal expansion coefficient [5]. Nonetheless, the geometry of the tool tends to have largely impact the final metallurgical properties of the processed material which can dictate part performance [6,8]. As the pin is plunged into the substrate, the processed region experiences localized frictional heating, thus softening the material and refining various material properties, which is typically a consequence of the grain refinement from the plasticized metal [9,10].

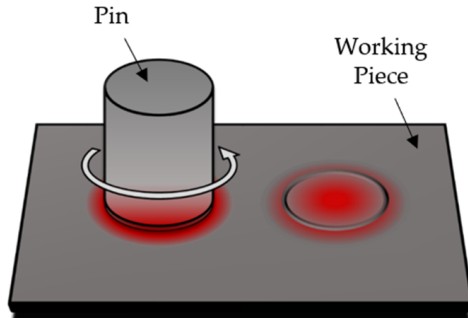

**Figure 1.** A schematic of a performed FSP process. In this figure, the rotating pin is plunging into a singular area of the working piece.

Taking a macroscopic view of FSP, surface process techniques such as plasma spraying and high-velocity oxy-fuel coatings [11,12] tend to pale in comparison to this process due to various distinct advantages. These advantages span from its technical flexibility, energy efficiency, environmental impacts, and of course, its ability for metallurgical optimization [3,13]. The metallurgical benefits are primarily related to the grain refining nature of FSP, which acts as the primary studied mechanism of FSP [14]. However, it has been noted that the heating from the processed zone tends to densify the surface, which can enhance the mechanical, corrosive, and tribological properties of the substrate [15]. It is particularly beneficial for the components fabricated using additive manufacturing (AM) techniques which typically suffer from microscopic porosity [16]. Given the popularity in additive based technologies [17], the study of FSP on components by AM techniques such as selective laser melting (SLM) [18], cold-spraying (CS) [19], and electron beam melting (EBM) [20] has been progressively increasing throughout the years. In fact, this increase in popularity has inspired many to use FSP as an additive manufacturing technology itself, which is commonly referred to as friction stir additive manufacturing (FSAM) [21]. Although a relatively new field of research, FSAM is proposed to be a better alternative to other AM techniques due to its conservation in production cost while maximizing the scale of production and its structural performance [22]. In essence, this technique is relatively simple where plates are continually added on top of each other and lap welded [23]. Acting as a quick and easy technique, this sector of research is one of the upcoming topics that should be observed closely in the coming years.

On the same note, for non-additive based FSP, many utilize this technique due to its great repeatability and dimensional stability when used. With the process being a one-step technique, it allows for ease of automation [24]. There is a low energy consumption rate with this process as the material enhancing mechanisms are due to frictional and plastic deformation. Lastly, this process also yields the benefits of eliminating the use of solvents, noise reduction, and mitigating the by-product of fumes, opposed to other processing methods which enable it as an appropriate choice for a green and efficient process [13]. This is primarily related to its simple singular step process (Figure 1), which can be easily programmed and replicated. On an industrial scale, it is predicted that FSP can result up to 95% reduction of energy used in heat treatment processes by improving the mechanical, tribological, corrosion, erosion, and general surface properties [25]. By utilizing these advantages, FSP has gained much traction since its introduction in industries such as military, naval, aerospace, automotive, and computing [26].

From an application perspective, FSP can be very useful as a post-surface processing technique. For example, explosive welding (EW) is a widely used fusion technique where metal plates can be cladded together in one singular step via instantaneous explosions. Although useful from a time efficiency timepoint, the intense temperature and pressure shockwave from the explosion result with surface defects such as cracks, holes, and voids [27]. Considering that large scale vessels and offshore structures largely employ EW, the usefulness of FSP is evident as it can easily repair their surfaces multiple times

throughout their lifetimes. Of course, other industries such as chemical processing equipment and refineries utilize also exhibit poor surface qualities from EW can also see benefits from FSP [28]. Considering the number of publications in this subject, there has been a drastic increase overtime with the number of publications increasing from 16,800 in 2010 to 93,800 in 2020 demonstrating the popularity of this processing technique [29].

The most common practice to perform FSP is rotating the specialized tool and moving it along a single linear direction along the workpiece [30]. However, this method is one of many that scientists use to investigate FSP operations. When using this process, there are numerous variables (aside from path direction) the operator can manipulate in order to refine the material. Such variables can be categorized into six broad categories based on the current state of literature. These categories consist of (1) machine variables, (2) tool design variables, (3) material properties, (4) number of passes, (5) active cooling techniques, and (6) particle reinforcement, as shown in Figure 2 [1,10,31]. When considering machine variables, there are a total of 8 different parameters which are typically altered in FSP. In general, these machining variables tend to impact the material flow of the process as well as the heat distribution of the substrate. It can be generalized that for higher melting point materials, the heat input should be higher, which can be manipulated by variables such as greater rotational speeds or longer processing times. In contrast, lower melting point materials should have the opposite [32]. In addition, other variables come into play, such as tooling geometries, the number of passes, and cooling techniques, which can enable a greater strain rate due to super-plasticity [33].

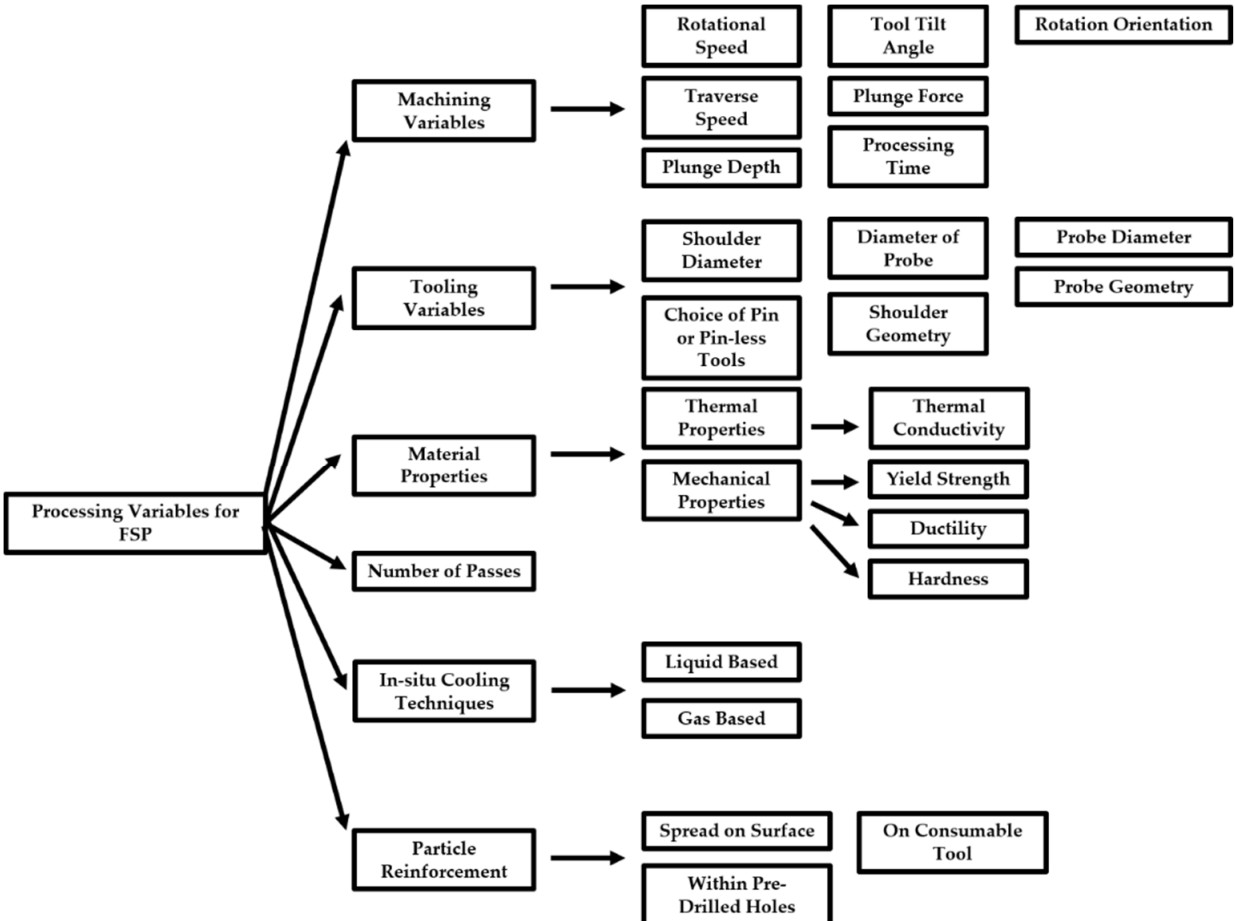

**Figure 2.** A detailed depiction of the effects of different processing variables on FSP steels.

## 2. Microstructural Mechanisms of Friction Stir Processing

In a broad sense, the manipulation of the previously discussed variables tends to largely affect the mechanical, microstructural, corrosion, friction, and wear properties of various materials [10,34–36]. However, these properties are tailored by the refinement of the grain [37]. This is achieved by a microstructure altering mechanism called dynamic recrystallization (DRX). It should also be mentioned that DRX can be segmented into four distinct sub-mechanisms called continuous DRX (CDRX), discontinuous DRX (DDRX), twinning induced dynamic recrystallization (TDRX), and geometric dynamic recrystallization (GDRX), depending on the material type [38,39]. In order to induce DRX, a strain-induced deformation must occur to the working piece [40]. Depending on the amount of strain exhibit on the material, atoms within the working piece are dislocated, thus allowing for the nucleation of new grains [41]. As these grains become re-nucleated, finer grain sizes are created, thus enhancing the microstructural properties of the material [42]. The general concept of DRX via FSP processes is shown in Figure 3. Additionally, to better visual the microstructural changes from FSP, electron backscatter diffraction (EBSD) images of dual-phase bimodal steel are also provided in Figure 4 [43].

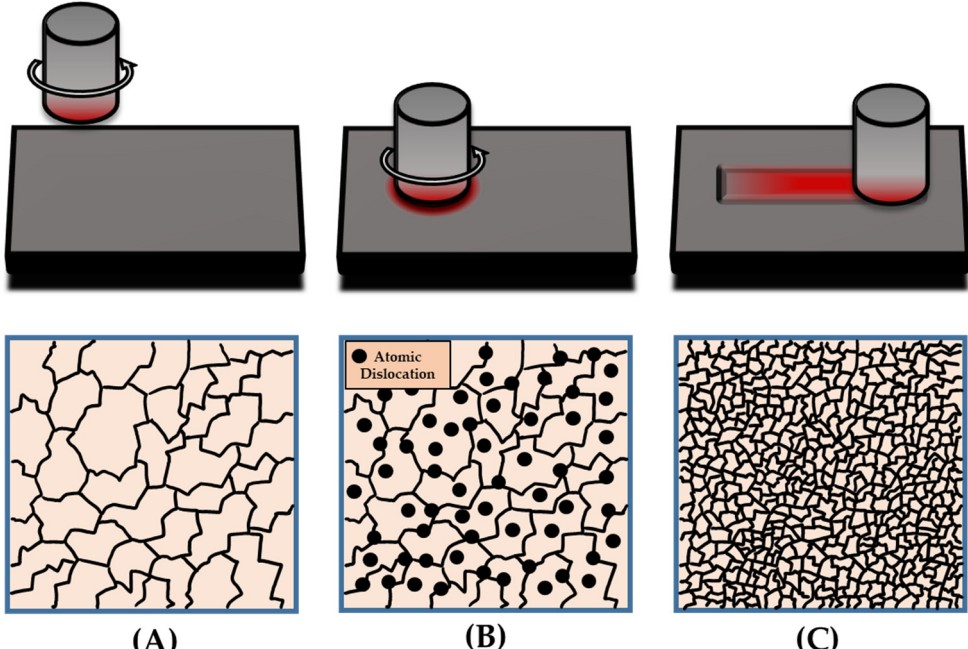

**Figure 3.** A schematic of the dynamic recrystallization process consisting of the (**A**) material in its base form, (**B**) the initial plunge into the working material, thus inducing severe atomic dislocations (shown by circles), (**C**) the reorientation of the atomic structure and the finalized grain structure.

Further exploring these mechanisms, CRDX is enabled by increasing dislocation slips due to thermal and strain-induced operations. As this happens, the lattice structure rotates with the sub-grain boundaries becoming high angled and, in consequence, more refined [44–46]. In contrast to this, DDRX is enabled by local elongation of the base grain or twin boundaries. During this migration, nucleation of the grain boundaries is initiated, thus resulting in grain refinement [39]. GDRX has been noted to occur during severe plastic deformation where grains impinge roughly two times the original grain size [47]. The most novel mechanism to date, however, is TDRX. Namely, this occurrence differs from the other stated mechanisms in the aspect that multiple processes can enable this recrystallization process. One process is related to the strain-induced transformation of high angle grain boundaries (HABG) from original twin boundaries (TB). In some instances, TBs will slip and migrate due to the annealing effect of frictional heating. This will result in the grains re-nucleating into equiaxed morphologies [45]. However, it should be noted that

regardless of the mechanism, the grain size will be refined, thus improving the properties of the material.

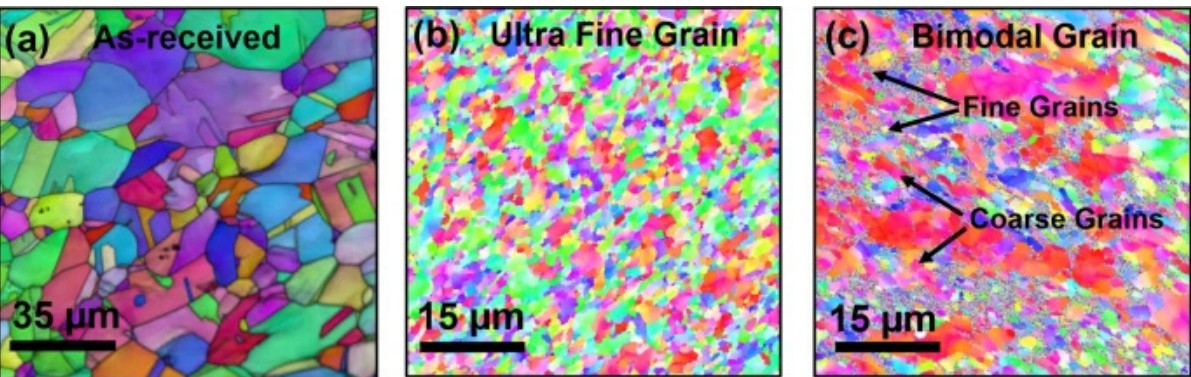

**Figure 4.** A representation of the ultrafine microstructural refinement of dual-phase bimodal steel (**a**) before and (**b**–**c**) after FSP, reprinted with permission from Arora et al. [43], under CC BY 4.0; published by Scientific Reports, 2019.

Post FSP, the processed zone can be categorized into four distinct regions. These regions consist of the stir region (SR), thermo-mechanical affected zone (TMAZ), heat affected zone (HAZ), and the base material (BM) [7]. Figure 5 demonstrates this with respect to the utilized tool in-vivo. What differentiates these zones from each other is the general change of grain size away from the area of contact [44]. Through a direct impact of shear stress from the tool as well as the frictional heat, the area of contact is identified as the BM. This region will have the finest grain size in comparison to all of the other zones. Moving directly out of the HAZ is the TMAZ. This area is greatly impacted by processing, thus resulting in finer grains. Outside of this region then exists the HAZ, where the frictional heat still impacts the sample. As the grains begin to increase in size, the last region of the worked piece is identified as the base metal (BM) where its properties are unaffected by the FSP operation [48].

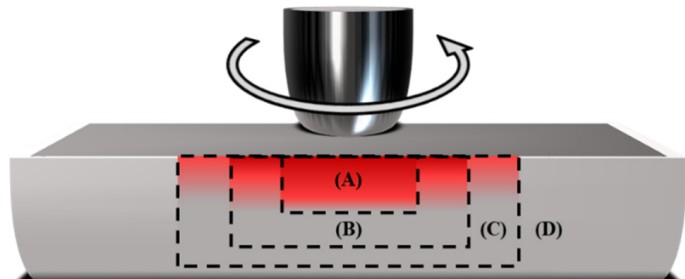

**Figure 5.** A depiction of the general processing zones derived from FSP. Zone A depicts the stir region (SR). Zone B is the thermomechanical affected region (TMAZ). Zone C is the heat affected zone (HAZ) and Zone D is the base material (BM).

Understanding the basics of FSP, a vast amount of literature focuses on the processing of soft-based metals due to their structural uses in industries such as aerospace and transportation [49]. Considering that softer metals such as aluminum or magnesium have a relatively low toughness high ductility, the high strain rate induced from processing can result with impactful improvements to their metallurgical properties [50]. In contrast to this, harder metals such as titanium, nickel, cobalt, and steel are often overlooked in research compared to softer metals due to their high melting points [51]. However, the growth of scientific knowledge in this topic has enabled new insights on how tougher materials can be improved from this refining process. In particular, steel is one of the most widely industrialized materials in this list being used in various industrial sectors such as transportation, aerospace, and medicine [52–54]. However, due to the intrinsic



limitations of steel, there is an increasing need to enhance its metallurgical properties, thus expanding its long-term usability. Typically the primary concerns are found from excessive material degradation as well as corrosion over long periods of time [55]. In fact, several studies found that a total of $2.5 trillion is lost annually due to corrosion [56,57]. The primary industries which are affected can span from transportation, to infrastructure and manufacturing. Keeping this in mind, large sums of money can be lost overtime due to this which generates an increasing need for its optimization.

Understanding the economic impacts of excessive material degradation, this review will specifically focus on the advancements of knowledge of FSP for steel and how its tribological and corrosion-based properties are enhanced due to the microstructural refinement and material densification from FSP [58]. Although there exists a basic literature review regarding the impacts of FSW for steel [59], this review differs in the aspect that we will be focusing on the processing of steel rather than the joining of it. This, in turn, creates a larger discussion of tribological and corrosion responses from processing compared to the weld quality characteristics found from FSW. From this review, we hope to establish a valuable source for researchers in this field to refer to when pursuing this research with insightful discussions of the future developments and voids that should be addressed in this field of research [59].

## 3. Mechanical and Tribological Enhancements

When considering mechanical and tribological properties, it is widely known that the grain structure influences their performances as per the Hall–Petch relationship,

$$H = H_o + k_H d^{-1/2} \tag{1}$$

The relationship states that the grain size is inversely proportional to the mechanical strength [60], which has also been observed for steel processed by FSP [61,62]. In fact, many of the microstructural refinements by FSP have been reported to span from the nano to micro-meter range [63], which depends not only on the deformation process but also on heat induced and heat transfer [64]. By obtaining a better understanding of how the processing parameters influence the microstructure of FSP steel, mechanical and tribological properties can be further enhanced. For reference, Table 1 will summarize the findings of this section.

**Table 1.** Publications relating to the microstructural, mechanical, and tribological refinement of FSP steels.

| Article Name | Summarization | Reference |
|---|---|---|
| Microstructure and Mechanical Properties of Friction Stir Processed AISI 316L Stainless Steel | • FSP was employed on AISI 316L steel sheets<br>• Lower heat input from lower rotational speeds enabled microstructural refinement<br>• Discontinuous dynamic recrystallization was the dominant mechanism of refined grains | [44] |
| On the Microstructure Evolution in Friction Stir Processed 2507 Super Duplex Stainless Steel and its Effect on Tensile Behaviour at Ambient and Elevated Temperatures | • FSP of 2507 super duplex stainless steel was studied<br>• Traverse speeds of 10, 25, 50, 100, 150, and 175 mm/min were studied<br>• Optimized grain size was found at 100 mm/min due to maximized strain and minimized heat input per unit area. | [65] |
| Relationship between microstructure and mechanical properties of friction stir processed AISI 316L steel produced by selective laser melting | • Additive manufactured (AM) 316L steel fabricated by selective laser melting (SLM) was subjected to FSP<br>• Micropores and microcracks from SLM were reduced from frictional heating<br>• Increasing dislocation density was observed | [66] |

**Table 1.** *Cont.*

| Article Name | Summarization | Reference |
|---|---|---|
| A novel approach to achieve high yield strength high nitrogen stainless steel with superior ductility and corrosion resistance | • FSP was employed using pin-less cermet tool with in-vivo water cooling<br>• Transient zones to the coarse grain zones were main contributors to tensile straight improvement<br>• Strain transfer relived localized strain concentrations | [67] |
| High Tensile Ductility and Strength in Dual-phase Bimodal Steel through Stationary Friction Stir Processing | • FSP was employed on dual-phase bimodal 316L steel<br>• Processing parameters consisted of moving pin and pin in static position for 15 min. Both tests were submerged in liquid solution.<br>• Work hardening and martensite phase transformation was noted for both substrates | [43] |
| Enhanced mechanical properties of medium carbon steel casting via friction stir processing and subsequent annealing | • Carbon steel castings were subjected to submerged FSP<br>• Preprocessed steel was subjected to annealing and quenching to improve mechanical properties<br>• Martensite grain structure was observed. Increased presence of carbide particles was also observed. | [68] |
| Friction stir processing of 316L stainless steel plate | • FSP was employed to 316L SS plates<br>• Defect free zone was found with low rotational and traveling speeds<br>• Formation of sigma phases was noted in this work. Densification of the substrate was also observed. | [69] |
| Tribological enhancement of AISI 420 martensitic stainless steel through friction-stir processing | • FSP was employed on AISI 430 martensitic stainless steel<br>• Increased wear resistance was observed due to austenitic phase transformation to martensite-based phase<br>• Elimination of carbides assisted with wear resistance | [70] |
| Effect of tool pin design on the microstructural evolutions and tribological characteristics of friction stir processed structural steel | • Tool pin design of FSP was studied against ST14 structural steel<br>• Square pin resulted in higher amount of plastic deformation, thus refining the grains and improving tribological performance | [71] |
| Effect of friction stir processing on the tribological performance of high carbon steel | • Original pearlite phase from 1080 carbon steel was transformed to martensite phase from frictional heat<br>• Increased wear resistance was observed due to the phase transformation | [72] |
| Rapid surface hardening and enhanced tribological performance of 4140 steel by friction stir processing | • Martensite transformation on 4140 steel was observed due to FSP<br>• Surface hardening was observed, which assisted with tribological performance. | [73] |

### 3.1. Microstructural Changes

From a microstructural point of view, the grain refining mechanism of FSP is largely influenced from the amount of strain induced from the tool piece. Typically, the rotational speed of the tool piece as well as its translational velocity across the working piece are the most impactful parameters for microstructural refinement. To elucidate, Hajian et al. [44] provide an insightful view on the microstructural refinement of FSP for AISI 316L steel sheets. In this work, FSP was employed using the rotational speeds of 200 and 315 RPM at a constant traverse speed of 63 mm min$^{-1}$. Interestingly, both the processing speeds enabled a complete refinement of the microstructure despite the common expectation of grain refinement at high rotational speeds. In fact, the heat input was enough to allow for plastic flow of the material, thus creating a symmetric basin-like shape throughout the entirety

of the plate. This can be largely related to the high stacking fault energy (SFE) induced from the FSP process, which acts as a large contributor to recovery and recrystallization of single phased, cubic materials. Due to this, it was determined that discontinuous dynamic recrystallization occurred in the system. Mishra et al. [65] also observed similar behavior while focusing on the effects of FSP on 2507 super duplex stainless steel. The independent processing variables consisted of the rotational speed and the tilt angle, measured at 800 RPM and 0°. The loading force was set to 9 kN while an Ar-gas shielding was used in order to prevent oxidation. The investigated variable in this study was the traverse speed 10, 25, 50, 100, 150, and 175 mm/min. Demonstrated in Figure 6, the least grain size was observed when the traverse speed was 100 mm/min.

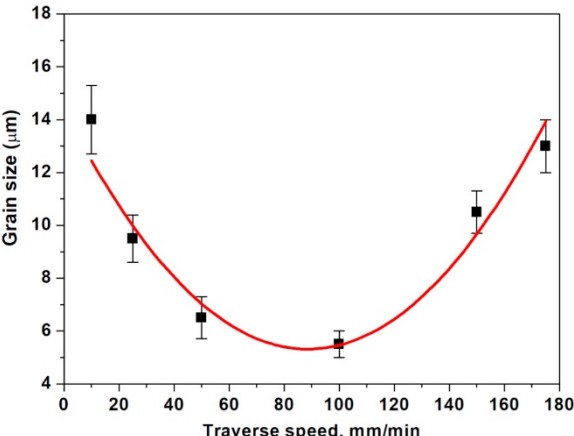

**Figure 6.** The relationship of grain size and traverse speed for FSP 2507 steel reprinted with permission from Mishra et al., 2018 [65].

To understand the observed phenomenon, the authors investigated two potential explanations based on existing literature. First, the relationship between traverse speed and rotational speed, also known as the $V/\omega$ ratio, was investigated. This concept was first investigated in FSW studies whether it was found that as this ratio increases, the grain size becomes more refined [74,75]. However, the optimal grain size was found at 100 mm/min, where any increasing traverse speeds afterward actually resulted in an increase in grain size. In order to explain this, the relationship of average heat input per unit area was investigated in regards to thermo-mechanical changes in the sample. The equation to calculate average heat input per unit area is shown as,

$$Q = \frac{4}{3}\pi^2 \frac{\mu P \omega R^3}{V} \qquad (2)$$

where $V$ is traverse speed, $\mu$ is the coefficient of friction, $P$ is the contact pressure, $\omega$ is the rotational speed, and $R$ is the radius of the shoulder of the tool. This is explained as the adiabatic buildup of heat in the sample actually negated the grain refinement effects of the $V/\omega$ ratio, thus explaining the parabolic curve in Figure 6 It is theorized that if the thickness of the sheet increases, there will be more material to absorb the frictional heating, thus allowing the grains to be further refined. Comparing these results to other related literature, they all yield similar results [44,76]. Understanding these mechanisms, the influence of grain refinement on the hardness, tensile, and tribological properties will be further discussed in the following sections.

### 3.2. Mechanical Properties

Based from the previous section, it can be insinuated that the refinement of steel microstructures due to sheer plastic deformation and frictional heating can improve their mechanical properties based on the Hall–Petch relationship [77].

Where the grain size is represented by d and the constants relating to the hardness of the material are shown by $H_o$ and $k_H$ [60]. The implications of increased steel hardness expand to numerous industries where plastic deformation is quite prevalent [78]. Examples of such industries span from energy, transportation, as well as construction [79]. With these industries greatly relying on efficient and durable steels, FSP poises as one processing solution that many can utilize to increase the strength of used steels in a cheap and effective manner.

In a recent study, Peng et al. [66] studied the effects of FSP on select laser melted (SLM) AISI 316L steel. The additive manufactured (AM) SLM part was used as the dependent variable in this study as one singular test was conducted. The rotational and processing tool speeds tested were at 375 RPM and 37.5 mm/min, with the tool tilt being set to 2°. As expected, the nature of FSP enabled the densification of the SLM substrate reducing the amount of micropores and microcracks along the molten pool region of the lasered region, as shown in Figure 7. As indicated from EBSD measurements, the amount of low angle grain boundaries (misorientation angles between grains = 2–15°) reduce indicating increased dislocation density from the DRX mechanisms. This implies that the formation of new grains is taking place due to the accumulation of sub-grain boundaries from the continuous dislocation buildup from the high-strain rate of FSP. TEM measurements later confirmed these measurements. Consequently, the nano-hardness of the sample was increased from 3.61 GPa to 4.18 GPa at the stir-zone of the sample. This was also shown with the micro-hardness of the samples as it was improved from 218 HV to 272 HV. The tensile and yield strength were also increased by 18% and 29%, respectively, which can be largely attributed to the refined grains and densified nature of the substrate. Of course, work hardening from FSP is also another factor that should be considered with these improvements.

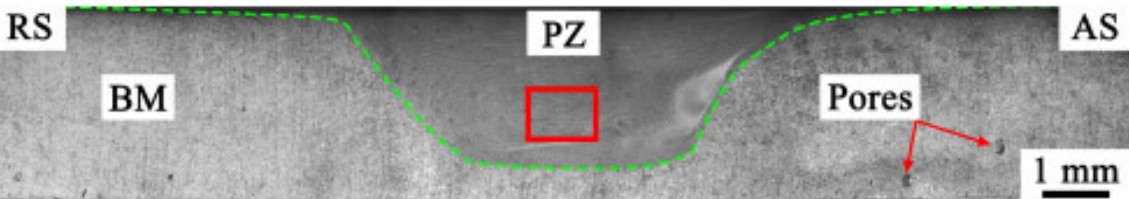

**Figure 7.** A cross-sectional view of the distribution of pores of SLM 316L where RS is the retreating side, AS is the advancing side, PZ is the processed zone, and BM is the base material. This figure was reprinted with the permission of Peng et al., 2020 [66].

Understanding the trends of the discussed works shows that low rotational speeds tend to increase mechanical hardness. However, in order to obtain a better grasp of the relationship between rotational speed and hardness, various studies on steels were observed and plotted in Figure 8. Of course, small variations in the initial processing methods will be different, however, this trend will help demonstrate the general enhancements of both rotational speed and hardness throughout steel samples.

As depicted, the decrease in tool rotational speed results in a higher hardness value for FSP 316L steel samples. Although there might be variations in other processing parameters, it is important to see an inversely proportional trend to tool rotational speed and hardness. An explanation for this can be described by the amount of frictional heating being displaced by the rotating tool. As the tool speed increases, greater localized heating occurs, thus disrupting the dynamic recrystallization process that occurs in vivo of the processing operation. With less rotational speed, grains are redefined, thus resulting in an increase in hardness based on the previously discussed Hall–Petch relationship [66].

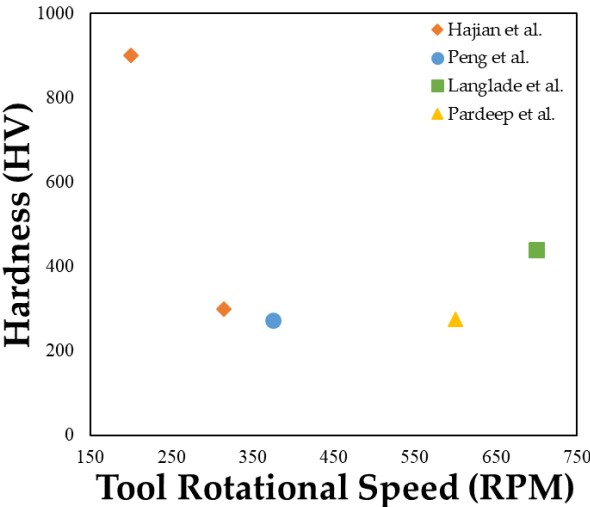

**Figure 8.** Relationship between hardness and tool rotational speed adapted from the work of Hajian et al., Peng et al., Langlade et al. and Pardeep et al. [44,66,80,81].

One commonality that is also observed in these works is related to the discussion of the heating distribution as well as the strain rates of FSP in regards to the tool rotational speed. Interestingly, the hardness appears to be the highest at a rotational speed of 200 RPM where the hardness gradually decreases until 700 RPM, somewhat resembling a parabolic curve. To explain this trend, it is important to evaluate the impacts of rotational speed and heat generation in relation to strain rate. Typically, the rotational speed tends to share a relationship with strain rate, which is to be expected due to the nature of FSP, which in turn results in elongated grains, thus enabling dynamic recrystallization [82]. Likewise, the strain rate can also be lowered due to the heat generation of the tool piece [83,84]. In this case, the lowest rotational speed enabled to highest strain rates while minimizing heat generation. Theoretically, with an increase in rotational speed, there will be a greater strain, however considering that up to 95% of the heat generated from the frictional stirring is constrained to the working material [85], it can be implied that the amount of heat generation far exceeds the strain rate at higher tool rotations. This seems to be the case until 700 RPM, which shows a slight increase in hardness. Although this might be counterintuitive, this increase in hardness can be explained by the increase in traverse speed, being recorded at 200 mm min$^{-1}$ [80] compared to the other recorded values measuring at most 100 mm min$^{-1}$. [44,66,81]. With decreasing traverse speeds, the peak temperature of the stirring increases which influences the final grain size [86].

Equally as important as hardness properties, tensile properties are also one feature that is enhanced via FSP. Enhancements to tensile strength have been greatly noted in a wide array of studies commonly featuring some form of soft metals such as aluminum or magnesium [87–90]. However, the quantity of studies involving steel is marginal compared to these softer metals. Given that steel is one of the top performing metals in terms of tensile strength [91], further improving these properties would allow for a wider array of applications for which steel can be used [92]. In this sense, industries that typically rely on large construction models or involve a lot of dynamic work will greatly benefit from increases in steel tensile strength [93].

One recent publication investigates this by evaluating the specific tensile strength enhancements of high nitrogen stainless steel via FSP [67]. In this study, a pin-less cermet tool was utilized with a rotational speed of 300 RPM and a traverse speed of 25 mm/min while being cooled with water in-vivo. Consequently, it was recorded that the steel improved from ~540 MPa to ~950 MPa at 30% elongation. The authors attribute this greatly to the grain structure formed during the FSP process. Specifically, the transient zones to the coarse grain zones were the main contributors to the tensile strength improvement as the strain transfer from the connecting regions allowed the material to relieve localized

strain concentrations during testing. Figure 9 demonstrates the transient and coarse grain zones through disorientation maps at 5%, 10%, and 20% tensile strains.

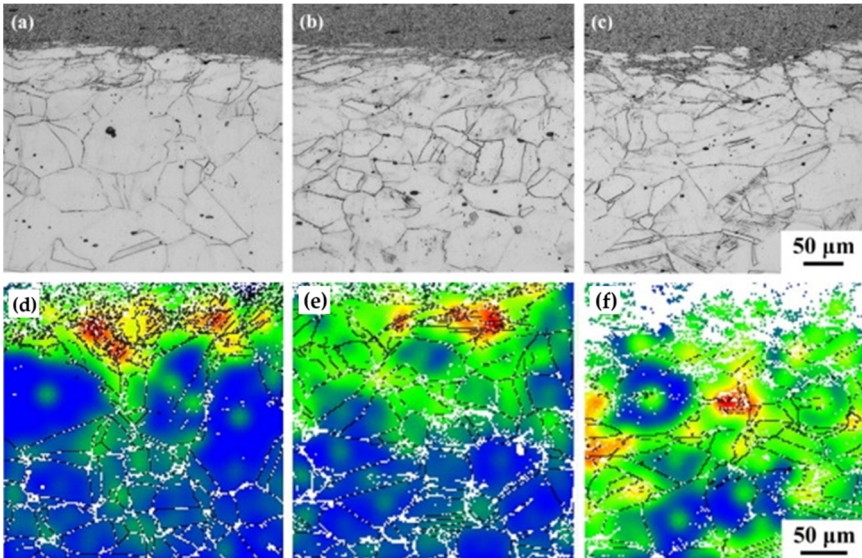

**Figure 9.** The transfer of transient zones to coarse grain zones of FSP high nitrogen stainless steel at tensile strains (**a**,**d**) 5%, (**b**,**e**) 10%, and (**c**,**f**) 20%. This figure was reprinted with permission of Zhang et al., 2019 [67].

Likewise, Arora et al. [43] also came to similar conclusions as dual-phase bimodal 316L steel was subjugated to FSP. As expected, the grains were refined in this process to a 500 nm size from the starting 10 µm starting size. In order to achieve these results, two different processing parameters were used. Both were keeping a rotational speed of 1800 RPM, one test was conducted using a traverse speed of 20 mm/min, whereas the second test was kept at a static position for 15 min. In both tests, the samples were submerged in a liquid solution to mitigate excessive frictional heating. The liquid solution contents were composed of a 50–50 mixture of ethanol and distilled water. Given the decrease in grain size, both tested samples were drastically improved over the base material, which measured at 300 MPa. In the case of the dynamic processing sample, the yield strength was measured at 450 MPa, falling behind the 620 MPa recording the static test produced. Based on grain analysis, it was determined that there was a larger quantity of martensite grains in the coarse austenitic phase of the samples. Consequently, significant work-hardening occurred during tensile testing, thus suspending the plastic deformation typically induced during testing.

Xue et al. [68] conducted a similar study with carbon steel castings subjected to submerged FSP. Preprocessing of the steel was performed in terms of annealing and quenching in order further to improve the mechanical properties of the base material. More specifically, a 920 °C annealing temperature was used for a duration of 12 h. After being quenched, the BM was annealed again at 620 °C for 12 h to further improve the mechanical properties of the sample. Three samples were fabricated with a rotational and traverse speed of 400 RPM and 50 mm/min. One was kept as is, and the other two were again annealed for 2 h at both 500 and 600 °C. Results indicate that the yield strength of the non-post heat-treated sample had the greatest yield strength, measuring at 2070 MPa compared to the 520 MPa measurement for the base sample. Improvements were also shown for the post-annealed samples, however, they measured at the values of 1270 MPa for the 500 °C sample, and 925 MPa for the 600 °C sample. Aside from the refined grains, the authors contribute these changes due to the greater presence of martensite-type grain structures in the FSP samples. This was most prevalent in the non-heat-treated sample where a greater presence of ferrite-based structures was found via SEM. The SEM micrographs also

indicated an increase in the amount of carbide particles within both annealed samples. It should be mentioned that the elongation percentage of the as processed FSP sample was 4%, much less than the 21.8% found in the base material. However, the 600 °C samples showed a marginal improvement to 23.4%.

In addition to the previous property enhancing improvements of FSP, one important outcomes that should be discussed of the optimization of pore distribution from FSP [94,95]. Although briefly discussed, the densification of material also has some impacts on the mechanical performance of steels through the high temperatures induced from frictional heating [96]. However, despite the amount of existing studies reviewing this property, no literature focuses on the porosity changing effects of FSP for steels. Only one study briefly mentions the change in porosity as one contributor to an enhancement in mechanical strength [69]. This could be attributed to a variety of factors, such as the general novelty of FSP or simply because softer metals tend to be more investigated. Regardless, this field of research could potentially yield impactful results and should be further explored in the future.

### 3.3. Tribological Properties

Enhancements in surface properties, or more commonly known as tribological properties, have been noted through the usage of FSP [97]. Studying the effects of friction, wear, and lubrication, the study of tribology has a tremendous impact on energy-conserving processes and mitigating material wear in contacting operations [98,99] which can be quite prevalent in industries such as automation [100]. Understanding that FSP is a direct surface processing technique in itself, the refinement of the material grain size directly influences the improvement of surface properties in any material, especially steels [73].

Dodds et al. [70] focus on these enhancements with AISI 420 martensitic stainless steel. In this study, four different samples were investigated to allow for comparison to the FSP sample. The specific preset conditions for the samples consisted of an as-received annealed sample, an as-received annealed sample with FSP, and, lastly, an as-quenched sample. The traverse and spindle speeds of the tool were set to 150 mm/min and 300 RPM with a 2° head tilt. Given this, one of the primary goals of this study was to enhance the wear resistance of the material due to the grain-refining nature of FSP. The processing conditions for the work consisted of a sliding speed of 5 mm/s, reciprocating frequency of 0.5 Hz, a normal load of 49.05 N, and a stroke length of 5 mm over a 3000 mm distance. The friction stirred regions of 1.5, 3, and 4.5 mm from the stir zone were studied when determining wear depths. It was found that all regions increased in wear resistance, with the 1.5 mm region increasing the most at a 57% wear reduction. The authors explain this result due to the many of the austenite-based phases transforming to the martensite-based phases. Of course, the microstructural refinement is closely related to these improvements, however, the elimination of carbides as well as the formation of a tribo-film acting as a preventive layer against intense abrasive wear. In fact, the micro-abrasion wear resistance constant in relation each materials hardness is shown in Figure 10. Additionally, the martensite present on the surface also helped increase plowing resistance, which in turn can create third body wear from the brittle fracture of the asperities.

Ameneh et al. [71] also conducted the tribological study in which the effects of the tool pin design were studied in ST14 structural steel. All being composed of tungsten carbide, the tool designs used for the study consisted of square, triangular, and conical geometries. FSP processing parameters were held constant, the traverse speed set to 100 mm/min, rotational speed at 400 RPM, and a 5 kN axial force. With the fabricated samples, sliding tests were conducted at 250 m travel intervals for a total of 1000 m. With this data, the coefficient of friction (COF), wear volumes and wear rates of the samples were investigated. It was found that the square pin yielded the most improved results out of all the tools. The authors attribute this to the higher amount of plastic deformation that occurred by the tool compared to a more rounded geometry such as the conical tool. This can also be explained

by the 85.3% grain reduction measured by the square tool in this study. Similar findings can also be found in the work of Rezaei-Nejad et al. [101].

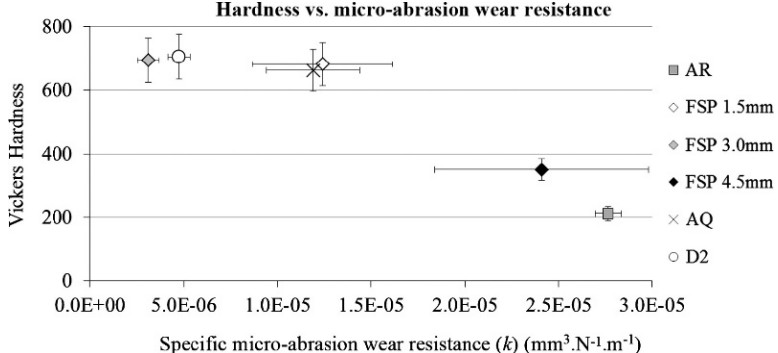

**Figure 10.** The relationship between wear and hardness for FSP AISI 420 martensitic stainless steel, reprinted with permission of Dodds et al., 2013 [70].

Aside from microstructural enhancements, the phase transformation induced from FSP can also result in surface hardening, implying improved tribological performance, as found in the work by Aldajah et al. [72]. In their work, 1080 carbon steel was subjected to FSP under a normal force of 5 kN while maintaining a constant rotational and translation speed of 1000 RPM and 15 mm/s. Having 8 total passes, 1 pass was done without any overlap, whereas each subsequent pass was overlapped by 20% under the assumption that the localized strain would be increased. Originally containing a pearlite phase, the heat generated from processing resulted exceeded the temperature range for the austenitic formation and instead resulted in a martensite transformation from the rapid cooling. As expected, the COF in dry sliding conditions decreased from 0.85 to 0.55 for both the single and multiple pass regions. Similar trends were observed with the wear rate, with the single-pass sample having the lowest wear rate out of all the tested specimens. Although the reasons for the single-pass sample are not detailed in the study, it can be hypothesized that the frictional heating generated from FSP was sufficient enough to allow for optimal recrystallization compared to the multi-pass sample.

Interestingly enough, Cinta and Oyelayo [73] also induced a martensite transformation of FSP steel which similarly resulted in surface hardening. In the case of their work, FSP was employed on 4140 steel using a processing depth of 5 mm, translation speed of 2 mm/s, rotational speed of 275 RPM, while keeping a constant force of 45 kN. Argon gas was additionally employed in order to mitigate any oxidizing defects. Unlike the work of Aldajah et al. [72], the COF was increased compared to the base sample. However, despite this finding, it was found that the FSP sample exhibited less wear for both the flat sample as well as the 52100-counter steel ball. Upon further observation, the wear mechanisms were quite different between the base material (BM) and the FSP sample. For the BM, a combination of abrasion and fatigue wear occurred on the wear track, with some transfer layer formation occurring sporadically along the wear track. Unlike to BM, the FSP specimen was mainly dominated by abrasion from the combination of microstructural refinement, martensite phase transformation, and a formation of a protective oxide layer increased carbon content. Although there are only four existing papers which cover this aspect of steel for FSP, it is suggested that additional research be done in order to fully understand the mechanisms for these findings. These studies suggest that the understanding of plastic deformation as well as the phase deformation by FSP can be used to process the steel surfaces for tribological applications.

## 4. Effects on Corrosion-Erosion Properties

Aside from the tribological and mechanical enhancements from FSP, improvement in corrosion and erosion resistance has also been reported due to three primary factors [102,103]. First is due to the densification of the substrate eliminating porosity defects, which can lead to severe pitting corrosion as well as increased corrosion rates from the increased surface area [103]. This is especially true in steel-based AM builds where the lack of fusion from the thermal heating process tends to result in a lack of fusion of particles, either resulting with balling from the Marangoni convection or the keyhole defect [17,104]. Secondly, the surface strengthening effect of FSP can enable a strong and robust passive oxide layer which can reduce the corrosion and erosion rate of processed samples [102]. Lastly, the third reason for enhanced corrosion resistance is simply due to the grain refining effects of FSP. Typically, smaller grain sizes tend to enhance the corrosion resistance due to the enrichment of Cr on the passive film [105–107]. Of course, it should be mentioned that the phase composition of steels also impacts their corrosion-resistant properties with austenite most corrosion-resistant phase [105]. In the current state of literature, there are various existing works that study the corrosion-erosion performance of FSP steels [70,102,108–113]. A brief summarization of these findings for reference is shown in Table 2.

**Table 2.** Publications relating to corrosion and erosion refinement of FSP steels.

| Article Name | Summarization | Reference |
|---|---|---|
| Enhancing the erosion-corrosion resistance of steel through friction stir processing | • Corrosion and erosion performance of 316L SS was studied after FSP<br>• Martensite transformation was observed for tested samples. Particularly lower rotational speeds enabled a greater transformation due to less frictional heating and greater strain.<br>• Finer grain stricture enabled faster passivation kinetics which assisted with corrosion and erosion performance. | [102] |
| Enhancement in hardness and corrosion resistance of AISI 420 martensitic stainless steel via friction stir processing | • FSP was employed to martensitic stainless steel<br>• Influence of FSP resulted in the γ phase being retained<br>• Densification of carbon atoms was observed thus enhancing pitting corrosion resistance | [111] |
| Corrosion behavior of friction stir processed AISI 430 ferritic stainless steel | • Corrosion behavior of AISI 430 ferritic stainless steel was studied post FSP.<br>• Corrosion resistance was diminished due to martensitic transformation<br>• Galvanic coupling was observed in martensite region | [113] |
| Friction Stir Processing of Stainless Steel for Ascertaining Its Superlative Performance in Bioimplant Applications | • FSP was employed to SS316L for biomedical applications<br>• Higher pitting resistance was observed due to refined grain size due to faster passivation kinetics<br>• Corrosion rate was attributed to faster Cr diffusion thus creating a more robust oxide layer. | [114] |
| Friction-stir processing of AISI 440C high-carbon martensitic stainless steel for improving hardness and corrosion resistance | • FSP was employed to AISI 440C high carbon martensitic stainless steel<br>• Dissolution of carbides during austenitic phase transformation allowed for a higher Cr content in the processed zone<br>• As a byproduct, the pitting resistance was enhanced | [115] |

**Table 2.** *Cont.*

| Article Name | Summarization | Reference |
|---|---|---|
| Improvement in cavitation erosion resistance of AISI 316L stainless steel by friction stir processing | • AISI 316L stainless steel was subjected to short term cavitation erosion resistance<br>• Over 30 h, cumulative mass loss and average depth from erosion decreased<br>• Dislocation buildup allowed for grater grain refinement which enabled a work hardening mechanisms from the shockwave pressure of cavitation erosion. | [116] |
| Improvement of cavitation erosion resistance of a duplex stainless steel through friction stir processing (FSP) | • UNS S32205 duplex stainless steel was subjected to FSP<br>• Microstructural refinement as well as a modified length of the $\alpha/\gamma$ interfaces was observed<br>• Refinement of austenitic/ferritic grains enabled work hardening thus reducing the total mass loss from cavitation erosion. | [117] |

In a recent study, Eskanadri et al. [113] studied the corrosion behavior of FSP AISI 430 ferritic stainless steel. The controlled variables in this work consisted of the tool tilt (3°) and rotational speed (1400 RPM) whereas the dependent variable was the travel speed which was altered between 16, 40 and 60 mm min$^{-1}$. One observation that was made with the travel speed was the formation of undesirable groove-like defects from insufficient heating. As a consequence, the sample with the greatest heat input (i.e., 16 mm min$^{-1}$) was chosen for investigation. Aside from the expected grain refining effects of FSP, an austenitic to martensitic transformation was observed. As a consequence, the corrosion resistance was diminished due to the enhanced corroding nature of martensite compared to ferrite which is due to the lower amount of Cr [118]. For comparison, the observed corrosion current of the bulk and FSP sample were shown to be at 900 and 1000 μA cm$^{-2}$, respectively. Galvanic coupling (i.e., the preferential corroding of one substrate when in contact with another substrate) was also observed in this work where the martensite regions were excessively corroded with increased pitting which can be largely attributed to the non-uniform passive layer of the martensite–ferrite matrix. It should also be taken into consideration that the residual stresses from processing also negatively impacted the corrosion rate of the steel.

Despite the lack of corrosive improvement of Eskandari's study, Selvam et al. [102] alternatively found improvements to SS316L steel both in erosion and corrosion from FSP. In this work, the rotational speeds of a pin-less milling machine were varied between 388 and 1800 RPM. Samples were immediately plunged in a subcooled coolant comprising of ethanol and distilled water. The sample was placed in a customized test rig for the erosion-corrosion tests that allowed for control over the impact velocity and distance between the sample and nozzle. With the tests lasting for 2 h, a mass flux rate of 0.5% sand in distilled water ran with the nozzle velocity set to 20 m/s. Static corrosion tests were also tested through cyclic polarization between −0.4 V to 1.5 V. Their experimental results indicate that the microstructure showed the greatest refinement of 0.67 μm at the 388 RPM speed compared to the 0.9 μm size found at 1800 RPM due to the greater strain rate and lesser temperature. From a phase point of view, a martensite transformation was observed at both rotational speeds. However, at the lower rotational speed there was a higher proportion of martensite contrast to the higher speed. Although it is typically known that higher rotational speeds enable a greater strain rate, the temperature from the high-strain rates can negatively influence the final grain size as per the Zener–Holloman parameter. Due to the increase in hardness from grain refinement, the mass loss due to erosion was reduced from 4.04 mg/h to 0.84 mg/h. However, one conflicting finding in this work compared to the previous study is that both the 388 and 1800 RPM samples exhibited greater corrosion resistance than the as-received sample. In this work, it was elucidated that the finer grain structure enabled an oxide layer with Fe$_2$O$_3$ as the outer

layer with $Cr_2O_3$ on the inner layer. This in turn, enabled faster passivation kinetics. From a pitting perspective, the 388 RPM sample exhibited worsened pitting resistance to the 1800 RPM solely due to its martensitic structure. These explanations as well as the fact that 430 SS tends to have a worsened corrosion performance due to the lack of chromium concentration [119].

Pan et al. [111] also found similar results where FSP was utilized on AISI 420 martensitic stainless steel. The dependent factors of this study pertained to the traverse speeds from the FSW machine. These speeds were varied from 150, 200, and 250 mm/min as the rotational speed of the pin-less tool was set to 2000 RPM. In this work, it was observed that the influence of FSP resulted in the retainment of the $\gamma$ phase as the second phase, thus resulting in undissolved carbides and densification of carbon atoms. This is in combination with the refinement of $M_{23}C_6$ enabled an enhancement of pitting corrosion resistance with $E_{corr} - E_{pit}$ increasing from 145 mV to 216 mV. This increase indicates the formation of a more robust protective oxide layer on the substrate surface which can act as a large factor for overall corrosion resistance. The same phenomenon was also noted in a separate more recent work by Pan et al. [115] where FSP was employed to AISI 440C high-carbon martensitic stainless steel. Having the same processing parameters as their previous study, it was found that the dissolution of carbides during the austenite phase transformation resulted in a higher Cr content, thus enhancing the pitting corrosion resistance of the samples.

As can be seen, the corrosion rate of FSP steels tends to largely differ due to the phase transformations induced from FSP. As a byproduct of this, the formation of protective oxide layers is enhanced, which have been shown to mainly improve the pitting performance of these substrates. Of course, all of these variables are dictated by the parameters of FSP, which are largely tied to the rotational speed. In order to visualize these differences, a collection of differences between the pitting and corrosion potentials are varied with respect to rotational speed and traverse speed. To simplify this investigation, the ratio of rotational speed to traverse speed will act as the comparing factor for the propensity of pitting corrosion. It should be mentioned that studies that contain all of the required factors (i.e., rotational speed, traverse speed, pitting potential, and corrosion potential) were fitted in this analysis. The results are depicted in Figure 11. As can be seen, a higher ratio of the rotational speed to traverse speed allows for the greatest pitting resistance. This observation can be easily explained due to the densification of the material as well as the greater propensity of an austenitic transformation compared to the martensite transformations observed with greater strain rates. Understanding this trend, it can be seen that for optimal pitting resistance, either the rotational speed should be increased or the traverse speeds should be decreased.

In addition to this, the cavitation erosion resistance of FSP steel has been studied in the literature. Hajian et al. [116] are one of many who have established a relation between the rotational speed parameters and its implication for short-term cavitation erosion resistance. In their work, AISI 316L stainless steel was subjected to FSP with the rotational speeds of 200 and 315 RPM with a constant traverse speed of 30 mm $min^{-1}$. Over a period of 30 h, both the cumulative mass loss and average depth from erosion of the FSP samples were drastically reduced. Unlike the previously discussed works, the authors explain the increased resistance to cavitation erosion due to the refinement of grain size rather than phase transformations which were not observed. Typically, low SFE materials also tend to have difficulty with dynamic recovery from the degree of separation of the partial dislocations. From FSP, there is a dislocation buildup along the grain boundaries, which can act as a work-hardening mechanism when subjected to the continuous shockwave impacts of cavitation erosion. The surface roughness of the various affected zones also indicates a decrease in average surface roughness from FSP, which follows suite the other results in this work.

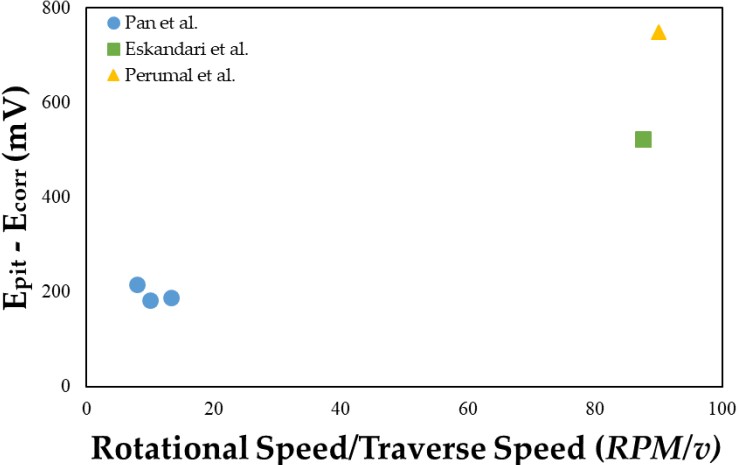

**Figure 11.** The pitting resistance of steel as a function of the ratio of rotational speed and traverse speed adapted from the work of Pan et al., Eskandari et al., and Permulal et al. [111,113,114].

These claims are also further supported from Escobar et al. [117] where UNS S32205 duplex stainless steel was similarly modified by FSP. However, in addition to the microstructural refinement, the length of the $\alpha/\gamma$ interfaces were also studied and related to the cavitation erosion resistance. Namely, from the equiaxed microstructure, the individual austenitic and ferritic grains are reduced thus minimizing the likelihood of crack propagation and the amount of dislocation movements induced from cavitation erosion. As this process occurs, the ferrite grains are continuously deformed until they reach their work hardening capacity. Meanwhile, the austenitic phase continuous to harden to the point where crack propagation induced from slip bands occur along the austenite and ferrite interface. It should be important to mention that in the work of Hajia et al. [116], there was no phase transformation. Rather, the original austenitic phase from the base-material remained the same after FSP. Additionally, both studies found that the 200 RPM rotational speed yielded superior results in comparison to the base materials. This can be indicative that the degree of plastic deformation to the heat generation is sufficient at lower rotational speeds, which acts as a good indicator for future studies in this field.

## 5. Applications of FSP Steel

Bearing in mind the feasibility of FSP for steel, it's important to highlight the applications of this research and the implications it has for day-to-day life. Namely, steel components that are exposed to high temperature applications tend to show the greatest benefits from FSP. This is especially true for the operation of chemical reactors, boilers, heat exchangers, or even water storage units in nuclear power plants [120]. As these components begin to age, the continual exposure of airborne chloride salts can influence the localized stresses to crack, thus resulting in early part failure. Instances of this have been observed from various nuclear facilities such as St. Lucie [121], Turkey Point [122], and San Onofre Nuclear Generating Station [123]. Given these structural failures, they may have serious implications for the general public, not to mention the large costs and time needed for reconstruction. Through FSP, reparation of inevitable surface defects is made possible, thus extending their longevity [120].

In a broader sense, other structural builds face similar issues as well. For example, fluid machineries tend to suffer from surface deterioration due to the continual impact of sand particles overtime, which is widely known as slurry erosion [124]. When considering other factors such as cavitation or corrosion, it's evident to how useful FSP can be for improving surface integrity. Alternatively, FSP can be employed to improve the tribological, corrosion, and cytocompatibility properties of bioimplants [114]. Through the formation of the passive layer, not only does the likelihood of pitting corrosion decrease, but the amount of metal ions released from the implant surface can be captured, thus reducing undesired

reactions with the surrounding biological fluid. Considering these points, it is evident to see the usefulness of FSP for real-world applications and the need for continual research in this field.

## 6. Summary and Future Outlook

FSP is a plastic deformation-based technique that allows for various metallurgical enhancements. Being developed in 1999, FSP acts as an eco-friendly, energy conserving technique that has gained much traction in its 21-year development. In fact, research in FSP has been drastically increasing in recent years as novel methods have been developed to enhance the results of this surface processing technique. Despite the increase in attention, studies involving steel-based parts tend to be overlooked compared to softer metals such as aluminum or magnesium. This review encompasses all of the current literature regarding this topic, from enhancements to mechanical properties to enhancement in microstructural, tribological, and corrosive/erosive based properties. It was found that the primary factors that determine these properties are related to the amount of grain refinement, material densification, and phase transformations induced from the synergistic combination of frictional heat and severe plastic deformation. In the case of mechanical performance, lower rotational speeds enable the greatest refinement of the microstructure due to the high strain rates compared to the frictional heat generated. Given that the strain rates are high enough, martensite phase transformations have been observed, which in turn enhance the wear resistance by minimizing brittle fracturing and tribo-film formation. From a corrosion-erosion point of view, the phase transformation induced from FSP tends to have a greater impact on the film formation and densification of steel which can largely dictate its performance. Given that the ratio of rotational speed to traverse speed increased, it has been demonstrated that the propensity of pitting corrosion is drastically reduced. Understanding these key mechanisms will allow researchers to optimize processing parameters and allow for greater industrial use of these post-processed parts. This is especially true in operations where surfaces are largely exposed to contacting and material degrading environments. However, the lack of literature on this topic is a large holdback of the potential performance/cost enhancing effects of FSP, especially from a tribological point of view. Understanding this, there is a need for additional research which can further enhance the knowledge of this research, thus benefiting its widespread industrial use.

**Author Contributions:** A.M.R.: conceptualization, methodology, formal analysis, and writing—original draft preparation; P.L.M.: conceptualization, methodology, writing—review and editing, and supervision; A.K.K.: conceptualization, methodology, and writing—review and editing. All authors have read and agreed to the published version of the manuscript.

**Funding:** This research received no external funding.

**Data Availability Statement:** The data presented in this study are available on request from the corresponding author. The data are not publicly available due to the data not being our own.

**Conflicts of Interest:** The authors declare no conflict of interest.

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
