# Peer review of "Friction Stir Processing on the Tribological, Corrosion, and Erosion Properties of Steel: A Review"

_jmmp, doi:10.3390/jmmp5030097_

Round 1
Reviewer 1 Report
The authors presented a systematic review on the friction Stir operations applied in steel. More specifically, the study is concentrated on the effects at the tribological, mechanical, corrosion, and erosion properties of the procedure. The paper is well structured by presenting the introduction, followed by microstructural mechanisms of friction Stir process, a discussion on the mechanical and tribological improvements of the process, the influence on the part’s properties on erosion and corrosion, followed by future research directions and applications. The research topic is interesting and represents high value of current interest. It is an original contribution in this field. Finally, the article is well written, clearly and concisely and is easily understood.
There are some minor comments, however, need to be addressed to improve the work presentation before being published in the “Manufacturing and Materials Processing” journal.
- I would expect a paragraph concerning the employment of friction Stir processes on Additive Manufacturing technologies.
- Hall-Petch relationship equation should be documented at the first time it is referred in the manuscript (in Page 5, not in Page 8)
- Better quality in the following figures is required, if it is possible:
Figure 5, Figure 7, Figure 9 and Figure 10.
4. Many important articles are reviewed in the manuscript; however, a higher number of references is expected about the effects on the corrosion and erosion properties.
5. Finally, a minor improvement in English is required as there are a few errors in the manuscript.
Reviewer 2 Report
This paper described FSP on the tribological, corrosion and erosion properties of steel and cited huge amount of recent papers. The authors have a good understanding in this field. The microstructures and mechanical properties for the steel after FSP have described systematic. Several questions have been put forward for further revision.
- In the introduction, the authors should be better describe the effect of FSP. In addition to the use mentioned by the authors, the authors shoud add some other uses of FSP, for example, rapairing the explosive clad plate.
- Fig.2, there are two Tool tilt angle, this should be checked.
- Fig.3, the pictures should not be appropriate. Is should be better change the pictures to EBSD photos.
- In the conclusion part, the author said "From a corrosion-erosion point of view, the phase transformation induced from
FSP tends to have a greater impact on the film formation and densification of steel w[112]," What is the meaning for steel w?
